# Willingness of caregivers to donate a kidney to a patient with end-stage renal disease: Findings from four dialysis providing health facilities in Uganda

Hope Bunori[1], Jonathan Izudi[1,2,3]*, John Bosco Alege[1], Francis Bajunirwe[2]

1 Institute of Public Health and Management, Clarke International University, Kampala, Uganda,
2 Department of Community Health, Mbarara University of Science and Technology, Mbarara, Uganda,
3 Infectious Diseases Institute, Makerere University College of Health Sciences, Kampala, Uganda

* jonahzd@gmail.com

**Data Availability Statement:** All relevant data are within the paper and its Supporting Information files.

## Abstract

Most patients with end-stage renal disease (ESRD) benefit from a kidney transplant but there is limited information from developing countries like Uganda about the willingness of caregivers for patients with end-stage kidney disease to donate a kidney. In this cross-sectional study, we examined the magnitude and factors associated with the willingness of caregivers to donate a kidney to their patient with ESRD in Kampala, Uganda. The study was conducted at four health facilities that provide kidney dialysis in Kampala, Uganda. We used a structured questionnaire to interview caregivers for patients with ESRD. Caregivers who reported they would consider donating a kidney to a patient with ESRD were considered willing and the rest as unwilling. We summarized data using descriptive statistics and used an adjusted prevalence risk ratio (aPRR) from a generalized linear model to establish factors independently associated with willingness to donate.We enrolled 125 participants with a mean age of 32.3±9.8 years and found 68 (54.4%) participants were willing to donate a kidney for transplant. Willingness to donate a kidney was more likely among older caregivers namely those aged 25–34 years (aPRR, 1.15; 95% CI, 1.01–1.31) and ≥35 years (aPRR 1.16; 95% CI, 1.05–1.29) compared to those aged 18–24 years, females compared to males (aPRR, 1.30; 95% CI, 1.19–1.42), those with a positive attitude towards organ donation (aPRR, 1.24; 95% CI, 1.13–1.36), and when organ kidney donation was permissible by the participant's religious faith (aPRR, 1.11; 95% CI, 1.01–1.22). Conversely, willingness to donate a kidney was less likely when the family did not approve of kidney donation (aPRR, 0.80; 95% CI, 0.71–0.90). We concluded that more than half of caregivers to patients with ESRD are willing to donate a kidney for transplant. To improve the willingness of caregivers to patients with ESRD in donating a kidney, the social, religious, and personal barriers to kidney donation may need to be addressed.

**Funding:** The author(s) received no specific funding for this work.

**Competing interests:** The authors declare that they have no competing interests.

## Introduction

Between five and seven million people in the world live with end-stage renal disease or ESRD [1], one of the complications of chronic kidney disease and require either dialysis or kidney transplant [2]. A kidney transplant is the preferred treatment option because it significantly improves life expectancy and quality of life [3]. In Uganda, patients with ESRD who require a kidney transplant are largely managed by dialysis due to the limited capacity of the health system to perform kidney transplants [4]. A kidney transplant is feasible if there is a willing donor and a willing recipient. Prior studies indicate that patients with ESRD are willing to accept kidney transplants provided they receive information on kidney transplantation, and if they perceive that a living person can donate a kidney, and also if they are convinced that the transplant would improve the quality of life [5]. More than a quarter of patients with ESRD are willing to pay for a kidney transplant particularly those in the high wealth quintile and or those with social support from family members and friends [6].

Although patients with ESRD are willing to accept kidney transplants and pay for them, organ donation faces several social, economic, religious, and cultural barriers in several sub-Saharan African countries. For instance, fear of complications and lack of trust in the capacity of the healthcare system to successfully perform an organ transplant [7], potential complications following transplant surgeries, and financial constraints [5] remain significant barriers. In the general adult population, a study in Nigeria found that males, people with secondary or tertiary level of education, and those married or never married, and the Muslims were more willing to donate an organ. [8] Furthermore, a study conducted among Ethiopian medical students showed they had a positive attitude towards organ donation, and males were more willing to donate than females [9].

Although there is considerable literature about the willingness to donate an organ, no study has explored the subject in Uganda. In particular, there is very limited data regarding the willingness of caregivers to donate a kidney to their patients with ESRD. Recently, Uganda passed the human organ donation and tissue transplant bill 2020 which awaits assent into law. The bill provides for the designation of hospitals as transplant centers, the approval of human organ, tissue, and cell banks, and appropriate consent for purposes of a human organ, tissue and cells donations, and transportation in Uganda [10]. At the moment, no law governs organ donation in Uganda.

In this study, we examined the level and factors associated with willingness to donate a kidney for transplant among caregivers for patients with ESRD at health facilities that provide kidney dialysis in Kampala, Uganda.

## Methods

### Study design and setting

We conducted a cross-sectional study at the four major health facilities that provide kidney dialysis in Kampala, Uganda. These health facilities included Kiruddu National Referral Hospital, Norvik Hospital, Victoria Hospital Bukoto, and Nakasero Hospital. Kiruddu National Referral Hospital is a public hospital while the rest are private health facilities. Kiruddu Referral Hospital has the highest bed capacity (200 patients), followed by Norvik Hospital (100 patients), then Nakasero Hospital (80 patients), and lastly Victoria Hospital with 70 patients. These study sites were purposively selected because they are the major known sites that provide kidney dialysis. We adhered to the Strengthening the Reporting of Observational Studies in Epidemiology (STROBE) guideline in reporting the study findings [11]. Also, we adhered to relevant national/institutional guidelines and regulations.

## Subjects

Our study population consisted of caregivers for patients with ESRD on kidney dialysis. We consecutively enrolled consenting caregivers aged $\geq$18 years at the respective study sites during the study period. The sampling of caregivers was proportionate to the size of the number of dialysis admissions during the study period at the respective health facilities.

## Measurements and data collection

We collected data using a structured, pre-coded, researcher-administered study tool that consisted of both open and closed-ended questions. The study tool was forward and backward translated from the English language to the local language "Luganda" and then back to the English language. The primary outcome of interest was the willingness of caregivers to donate a kidney to a patient with ESRD, measured as a binary outcome, namely yes and no. We asked the caregivers the question: *How would you rank your willingness to donate a kidney to this patient*?

The responses were either "I am willing to donate my kidney" or "I am not willing to donate my kidney". Each of these responses was followed by a question regarding the rationale. The independent variables examined are individual factors like age, sex, level of education, marital status, religion, residence, and the relationship of the caregiver with the patient. We also asked the participants: whether their family or family members would allow them to donate a kidney, whether the caregiver had ever received information about kidney donation, whether he/she was aware of the requirements for kidney donation, and whether she/he had ever received information about kidney donation while in the hospital. Furthermore, we obtained data about caregivers' concerns such as whether one can live after kidney donation, individual religious requirements to obtain permission for organ donation, awareness of the potential benefits of kidney donation and the associated risks, and the attitude they had towards kidney donation. We also asked the participants about their belief in the capacity of the health system to safely perform kidney donation and transplantation, and whether the process would require financial support. Our questionnaire had a acceptable psychometric property, with a Cronbach's alpha of 70.1% suggesting a good reliability or internal consistency [12].

## Statistical issues

**Sample size estimation.** Sample size estimation was based on the one-sample proportion approach for an unknown outcome. There were 160 participants across the study sites as the population size for the finite population correction factor. We assumed a hypothesized percentage of outcome in the population as 50% with a margin of error of ±5% at 95% confidence level. We determined that 114 participants are needed based on Kish and Leslie formula [13]. When we adjusted the estimated sample size for finite population correction of proportions and an additional 10% non-response rate to the estimated sample size of 114, the overall sample size needed was 125 participants. The computation was as follows: no = $Z^2$p(1-p)/$d^2$, where no = unadjusted sample size, Z = Z-score at 95% confidence level, p = proportion of the outcome in the population and d = acceptable margin of error. Therefore, no = $1.96^2$ X 0.5(1–0.5)/$0.05^2$ = 384. Adjusting for finite population correction of proportions using the formula, n = no/[1+(no-1/N)], where n = adjusted sample size, no = 384 and N = 160 (the finite population), n = 384/[1+(384-1/160)] = 114. With 10% non-response rate (11 participants), the overall sample size equals 125.

**Data analysis.** In the univariate analysis, we computed percentages for categorical variables such as sex and means with standard deviation or medians with interquartile ranges for numerical data like age. In the bivariate analysis, we cross-tabulated categorical variables with

the outcome and assessed statistically significant differences using the Chi-square or Fisher's exact test. We used the Student's t-test to examine mean differences in the variables with numerical data. We considered variables with p<0.15 at the bivariate analysis, those significant from previous literature, and those that demonstrate a plausible social or biologic link with the outcome for inclusion in the multivariable analysis. We conducted the analysis using a generalized linear model with a log-link function, Poisson distribution, and robust standard errors to avoid violation of the assumptions for the Poisson regression model [14]. We expressed the unadjusted and adjusted results using prevalence risk ratio (PRR) and corresponding 95% confidence interval (CI). We chose PRR over the odds ratio because the outcome was frequent and in such circumstances, the odds ratio would overestimate the association [15]. We presented the adjusted PRR (aPRR) in a prevalence risk ratio plot. The analysis was performed in Stata version 15 and R programing language (Statistical software version 4.0.2). All the analyses assumed a 5% level of statistical significance.

## Quality control measures

The Research Assistants received training for three days about the study protocol, study tools, and responsible conduct of research. We pre-tested the questionnaire outside the study setting to identify questions that were difficult to answer, those that were inappropriately worded or required to be refined. We adjusted the final study tool based on the feedback from the pre-testing exercise. During data collection, the completed study tools were checked by the Research Assistants in real-time to identify unanswered questions so that immediate responses were obtained. We recruited a team leader who supervised the data collection and sampled the completed study tools on daily basis and cross-checked them for missing data. We used Epidata for data entry and incorporated quality control checks such as skips, alerts, and range and legal values.

## Ethical issues

All the participants gave a written or thumb-printed informed consent in case they were not able to write their names. Before obtaining consent, we explained the study purpose, potential benefits and risks, and the withdrawal process. We ensured that the participants were free to withdraw from the study if and when they wish to do so. This study was approved by the Clarke International University Research Ethics Committee (reference # CIUREC/0208).

## Results

### Demographic characteristics of the participants

We studied 125 participants with an overall mean age of 32.3±9.8 years, and 68 (54.4%) were females, 74 (59.2%) belonged to the Anglican religious faith, 84 (67.2%) had attained tertiary or university level of education, 64 (51.2%) were married, 111 (88.8%) lived in a rural area, and 97 (77.6%) had ever heard of kidney donation (Table 1).

### Comparison of participants' characteristics by the willingness to donate a kidney for transplant among patients with ESRD

Table 2 compares the characteristics of the participants stratified by a willingness to donate a kidney to patients with ESRD. We found that 68 (54.4%) participants were willing to donate a kidney to a patient with ESRD.

The highest proportion of participants who were willing to donate a kidney included: 25–34 years (45.6%), female (61.8%), of the Anglican religious faith (64.7%), had received a tertiary

**Table 1. Participants demographic characteristics.**

| Characteristics | Levels | Total (n = 125) |
|---|---|---|
| Age categories | 18–24 | 26 (20.8) |
| | 25–34 | 56 (44.8) |
| | 35 or older | 43 (34.4) |
| Age (mean (SD)) | | 32.3 (9.8) |
| Sex | Male | 68 (54.4) |
| | Female | 57 (45.6) |
| Religion | Catholic | 34 (27.2) |
| | Muslim | 17 (13.6) |
| | Anglicans | 74 (59.2) |
| Level of education | None/primary | 19 (15.2) |
| | Secondary | 22 (17.6) |
| | Tertiary/university | 84 (67.2) |
| Marital status | Single/never married | 61 (48.8) |
| | Married | 64 (51.2) |
| Residence | Urban | 111 (88.8) |
| | Rural | 14 (11.2) |
| Relationship between the patient and caregiver | First-degree relative | 75 (60.0) |
| | A friend | 17 (13.6) |
| | Other relatives | 33 (26.4) |
| Ever heard of kidney donation | Yes | 97 (77.6) |
| | No | 28 (22.4) |

or university level of education (72.1%), married (51.5%), and urban resident (85.3%), and first-degree relative (55.9%) among others. We observed statistically significant differences in willingness to donate a kidney by sex (p<0.001), the willingness of the potential donors family to permit kidney donation (p<0.001), religious acceptability of organ donation (p<0.001), and the individual's attitude towards organ donation (p<0.001). The other variables such as age, level of education, marital status, residence, ever heard about kidney donation, awareness about kidney donation requirements, and received health education about kidney donation did not demonstrate statistically significant differences.

## Reasons for and against willingness to donate a kidney among caregivers for a patient with ESRD

The participants gave several reasons for and against their willingness to donate a kidney (Table 3). The notable reasons for willingness included the notion that life is an important and special gift, sympathy for someone on dialysis, and identification with the patient with ESRD. Conversely, those unwilling to donate a kidney stated lack of confidence in the healthcare system, fear of loss of body integrity and death, negative family influences, fear of limited success of surgery, negative religious influences, and fear of losing employment due to prolonged hospitalization following surgery.

## Unadjusted analysis of factors associated with willingness to donate a kidney to a patient with ESRD

In the unadjusted analysis (Table 4), the variables associated with a willingness to donate a kidney included female sex compared to male sex (PRR, 1.26; 95% CI, 1.13–1.40), religion allows

**Table 2. Caregivers' characteristics by the willingness to donate a kidney for patients with ESRD.**

| Characteristics | Levels | Willing to donate a kidney for transplant | | P-value |
|---|---|---|---|---|
| | | No (n = 57) | Yes (n = 68) | |
| Age categories | 18–24 | 15 (26.3) | 11 (16.2) | 0.334 |
| | 25–34 | 25 (43.9) | 31 (45.6) | |
| | 35 or older | 17 (29.8) | 26 (38.2) | |
| Age (mean (SD)) | | 30.5 (8.2) | 33.9 (10.8) | 0.058 |
| Sex | Male | 42 (73.7) | 26 (38.2) | <0.001 |
| | Female | 15 (26.3) | 42 (61.8) | |
| Religion | Catholic | 20 (35.1) | 14 (20.6) | 0.193 |
| | Muslim | 7 (12.3) | 10 (14.7) | |
| | Anglicans | 30 (52.6) | 44 (64.7) | |
| Level of education | None/primary | 7 (12.3) | 12 (17.6) | 0.060 |
| | Secondary | 15 (26.3) | 7 (10.3) | |
| | Tertiary/university | 35 (61.4) | 49 (72.1) | |
| Marital status | Single/never married | 28 (49.1) | 33 (48.5) | 1.000 |
| | Married | 29 (50.9) | 35 (51.5) | |
| Residence | Urban | 53 (93.0) | 58 (85.3) | 0.256 |
| | Rural | 4 (7.0) | 10 (14.7) | |
| Relationship between the patient and caregiver | First degree relative | 37 (64.9) | 38 (55.9) | 0.586 |
| | A friend | 7 (12.3) | 10 (14.7) | |
| | Other relative | 13 (22.8) | 20 (29.4) | |
| Family does not permit kidney donation | No | 26 (45.6) | 58 (85.3) | <0.001 |
| | Yes | 31 (54.4) | 10 (14.7) | |
| Ever heard of kidney donation | Yes | 42 (73.7) | 55 (80.9) | 0.456 |
| | No | 15 (26.3) | 13 (19.1) | |
| Aware of donor requirements for kidney donation | No | 35 (61.4) | 33 (48.5) | 0.208 |
| | Yes | 22 (38.6) | 35 (51.5) | |
| Ever received health education about kidney donation and transplantation | Yes | 8 (14.0) | 19 (27.9) | 0.096 |
| | No | 49 (86.0) | 49 (72.1) | |
| Knows that person can stay alive after kidney donation | No | 20 (35.1) | 20 (29.4) | 0.628 |
| | Yes | 37 (64.9) | 48 (70.6) | |
| Religion allows organ donation | No | 41 (71.9) | 24 (35.3) | <0.001 |
| | Yes | 16 (28.1) | 44 (64.7) | |
| Aware of the benefits and risks associated with kidney donation | No | 26 (45.6) | 40 (58.8) | 0.196 |
| | Yes | 31 (54.4) | 28 (41.2) | |
| Attitudes towards organ donation | Negative | 42 (73.7) | 22 (32.4) | <0.001 |
| | Positive | 15 (26.3) | 46 (67.6) | |
| The Health system is capable of performing kidney donation and transplantation | No | 39 (68.4) | 41 (60.3) | 0.450 |
| | Yes | 18 (31.6) | 27 (39.7) | |
| Kidney donation needs a lot of financial support | Yes | 27 (47.4) | 28 (41.2) | 0.607 |
| | No | 30 (52.6) | 40 (58.8) | |

Note: Values are column percentages in the form n/N.

donation (PRR, 1.27; 95% CI, 1.14–1.41), and positive attitude towards organ donation (PRR, 1.31; 95% CI, 1.17–1.45). On the other hand, the variables associated with a lower likelihood of willingness to donate a kidney include a secondary level of education compared to primary or

**Table 3. Reasons for and against willingness to donate a kidney to a patient with ESRD.**

| Reasons for willingness (n = 67)[#] | Frequency (%) |
|---|---|
| Life is important and a special gift | 35 (52.2) |
| I sympathize with someone on dialysis | 14 (20.9) |
| Identification with recipient | 11 (16.4) |
| Motivated by advancement in scientific knowledge | 3 (4.5) |
| Family relations | 3 (4.5) |
| Desire to help | 1 (1.5) |
| **Reasons for unwillingness (n = 61)** [#] | |
| No confidence in the healthcare system | 18 (29.5) |
| Loss of body integrity | 10 (16.4) |
| Fear of death | 14 (23.0) |
| Negative family influence | 10 (16.4) |
| Fear of losing employment due to prolonged hospitalization | 3 (4.9) |
| Fear of limited success of the surgery | 4 (6.6) |
| Religious faith does not allow organ donation | 2 (3.3) |

Note: #: Multiple responses were permissible.

no level of education (PRR, 0.81; 95% CI, 0.66–0.99), family refusal (PRR, 0.74; 95% CI, 0.65–0.83), and lack of health education about kidney donation and transplantation (PRR, 088; 95% CI, 0.78–0.99).

Fig 1 displays the results for an adjusted prevalence risk ratio plot.

**Table 4. Results of unadjusted analysis of factors associated with willingness to donate a kidney for transplant to a patient with ESRD.**

| Characteristics | Levels | Unadjusted analysis | |
|---|---|---|---|
| | | PRR | 95% CI |
| Age categories | 18–24 | 1 | |
| | 25–34 | 1.09 | (0.93,1.28) |
| | 35 or older | 1.13 | (0.96,1.33) |
| Sex | Male | 1 | |
| | Female | 1.26*** | (1.13,1.40) |
| Level of education | None/ or primary | 1 | |
| | Secondary | 0.81* | (0.66,0.99) |
| | Tertiary/ or university | 0.97 | (0.84,1.13) |
| The family does not permit kidney donation | No | 1 | |
| | Yes | 0.74*** | (0.65,0.83) |
| Ever received health education about kidney donation and transplantation | Yes | 1 | |
| | No | 0.88* | (0.78,0.99) |
| Religion allows organ donation | No | 1 | |
| | Yes | 1.27*** | (1.14,1.41) |
| Attitudes towards organ donation | Negative | 1 | |
| | Positive | 1.31*** | (1.17,1.45) |

**Note**: 1) Prevalence risk ratios (PRR) are exponentiated coefficients at 95% confidence level; 2) 95% confidence intervals are in parentheses; 3)

* $p < 0.05$

** $p < 0.01$

*** $p < 0.001$ at 5% significance level; 4) PRR: Unadjusted PRR.

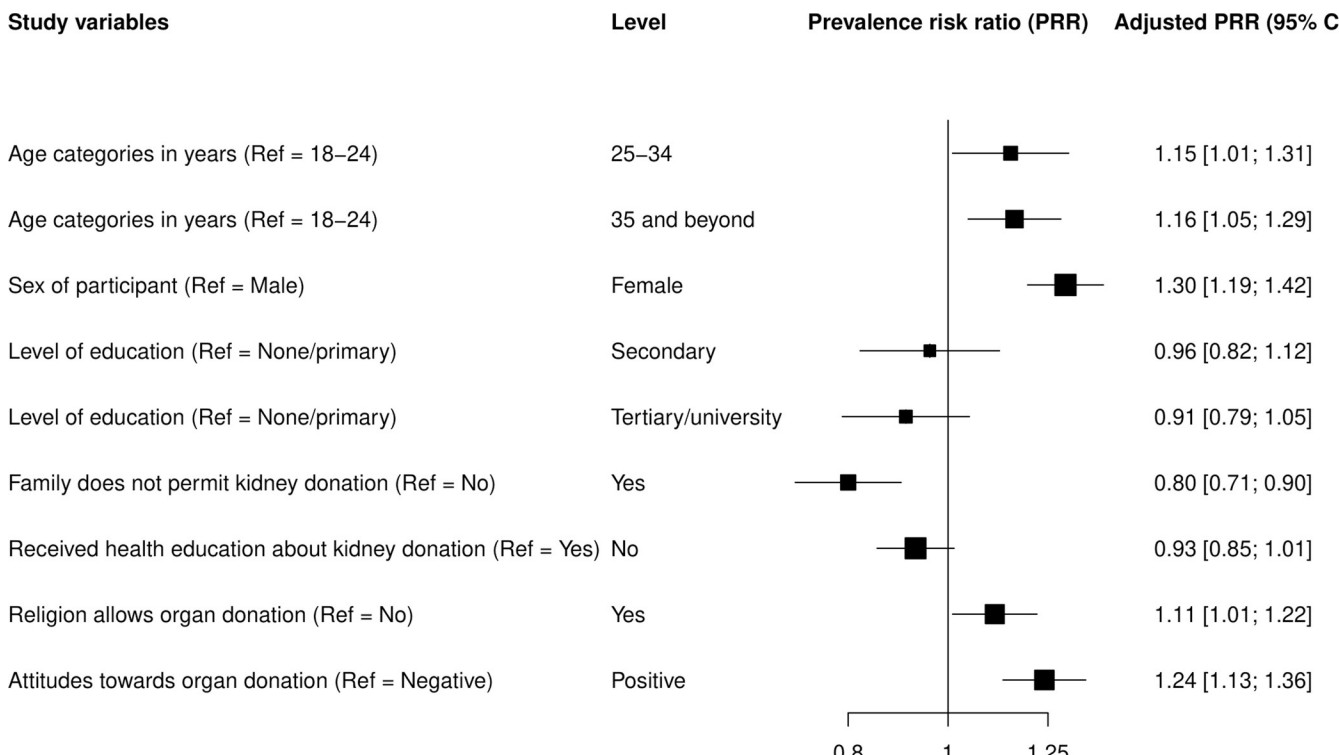

| Study variables | Level | Prevalence risk ratio (PRR) | Adjusted PRR (95% CI) |
|---|---|---|---|
| Age categories in years (Ref = 18–24) | 25–34 | | 1.15 [1.01; 1.31] |
| Age categories in years (Ref = 18–24) | 35 and beyond | | 1.16 [1.05; 1.29] |
| Sex of participant (Ref = Male) | Female | | 1.30 [1.19; 1.42] |
| Level of education (Ref = None/primary) | Secondary | | 0.96 [0.82; 1.12] |
| Level of education (Ref = None/primary) | Tertiary/university | | 0.91 [0.79; 1.05] |
| Family does not permit kidney donation (Ref = No) | Yes | | 0.80 [0.71; 0.90] |
| Received health education about kidney donation (Ref = Yes) | No | | 0.93 [0.85; 1.01] |
| Religion allows organ donation (Ref = No) | Yes | | 1.11 [1.01; 1.22] |
| Attitudes towards organ donation (Ref = Negative) | Positive | | 1.24 [1.13; 1.36] |

**Fig 1. Prevalence risk ratio plot showing results of the adjusted analysis.** Willingness to donate a kidney was more likely among participants aged 25–34 years (aPRR, 1.15; 95% CI, 1.01–1.31) and ≥35 years (aPRR, 1.16; 95% CI, 1.05–1.29) compared to 18–24 years. Females were 30% more likely to have the willingness to donate a kidney compared to males (aPRR, 1.30; 95% CI, 1.19–1.42). Willingness to donate a kidney was more likely if it was considered by the participant as allowable by their religion than when it was not permissible (aPRR, 1.11; 95% CI, 1.01–1.22). Participants who had a positive attitude toward organ donation were more likely to be willing to donate compared to those who had a negative attitude (aPRR, 1.24; 95% CI, 1.13–1.36). Willingness to donate a kidney was less likely if the family did not permit it (aPRR, 0.80; 95% CI, 0.71–0.90). Secondary (aPRR, 0.96; 95% CI, 0.82–1.12) and tertiary or university levels of education (aPRR, 0.91; 95% CI, 0.79–1.5) and receipt of health education about kidney donation and transplantation (aPRR, 0.93; 95% CI, 0.85–1.01) were not associated with willingness to donate a kidney.

## Discussion

We examined the willingness of caregivers to donate a kidney for transplant to their patient with ESRD and found more than half (54.4%) of them were willing and willingness was associated with age, sex, religion, attitude towards organ donation, and family approval. The proportion of those willing to donate a kidney was 51.4% among caregivers in China [16], and 48% in one community-based study in Ghana [17], and 48.5% in another study in the general population [18]. The proportion of willingness to donate a kidney in the present study is not far different when compared with that in these previous studies. In many settings, kidney donation or any other type of organ donation remains contentious. For example, one study conducted in Malaysia shows that people prefer to donate their organs after death than when alive [19]. The observed level of willingness to donate a kidney is suboptimal and this could be due to a lack of enabling legislation to regulate organ donation and transplantation. Currently, Uganda depends on the WHO guiding principles on human cells, tissue, and organ transplantation which limits potential donors to close relatives [20]. Therefore, the majority of the population is not knowledgeable about kidney or organ donation in general. Moreover, knowledge inadequacies about organ donation deter people from donating organs [21]. This is likely the case in most countries in sub-Saharan Africa.

In Uganda, kidney donation is only acceptable from matching family members. It is not clear whether the patients pay the donors as no formal regulations or legislation are in place. Family members may donate organs because it is culturally acceptable or tolerable compared to donations from other relatives and friends. Most of the participants interviewed highlighted a lack of confidence with the healthcare system's ability to successfully perform an organ donation and transplantation, fears concerning the loss of body image, and death following organ donation as reasons for their unwillingness alongside negative family and religious influences. There are limited studies that have been conducted in developing countries to compare our results with therefore more data are needed especially from qualitative studies to explore these findings in-depth.

Our finding that older persons are more willing to donate a kidney is consistent with a previous study in Singapore [22]. Another study shows that the relationship between age and organ donation is linear [23], suggesting age is an important social determinant of health [24]. Older persons may have more control in decision-making regarding their health and that of others compared to younger people. Older people may also comprehend complex and ethical issues surrounding organ donation much easier than young people. The interplay between age and kidney donation should be examined in further studies. However, we also note that caregivers are generally younger persons compared to the general population, probably due to the need to have an able-bodied person to attend to the needs of a very sick patient.

We found females are more willing to donate a kidney compared to males and this is consistent with a study conducted in Greece [25] but contrary to two studies in Nigeria [8,26] and another study in Saudia Arabia [27]. The differences might be explained by the sampling approach for the study populations and the measurement of outcomes of interest in the studies. The studies from Nigeria were conducted in the general population and the outcome of interest was a willingness to donate any body organ. Our study involved caregivers for patients with ESRD and examined their willingness to specifically donate a kidney. Also, inherent differences between males and females might explain the result. Females tend to be more compassionate and empathetic than males as highlighted in a past study [27].

Willingness to donate a kidney is more likely when one's religion permits it. The influence of religion on health matters is not new and remains controversial the world over. A recent systematic review reports that certain religious faiths encourage organ donation because it fitted well within the altruistic belief system whereas others maintain that organ donation is not permissible [28]. Even among participants in the same religious faith, there are diverse opinions regarding organ donation, with some accepting the practice while others have refused it completely [28]. We observed this phenomenon in this study. Our findings underscore a need for wider religious engagement in discussions about organ donation. Such discussions might improve the knowledge and acceptability of organ donation among religious leaders and their followers.

Our data show that participants with a positive attitude towards organ donation are more willing to donate a kidney, and this is consistent with the findings of a community-based cross-sectional study conducted in Karachi, Pakistan [21]. In our opinion, a positive attitude towards organ donation is an important step to catalyze willingness to donate or receive a kidney or any body organ. Fear of complications and mistrust of the health sector [7] and lack of sufficient information about organ donation are some of the notable reasons for unwillingness to donate any body organ(s). Some of these concerns emerge from the attitudes towards organ donation in general. Since attitude influences health behaviors and practices, there is a need to address attitudinal concerns about organ donation through mass health education.

Our data show that willingness to donate a kidney is less likely among participants in families that did not approve kidney donation consistent with findings of a past systematic review

[28]. The systematic review found that the family remarkably influences one's decision to either accept or refuse organ donation. This is because potential donors seek family approval before organ donation and when approval is not granted, their willingness to donate an organ wanes. Refusal of donated organs and organ donation by a family has been observed in the United Kingdom as well, with more than four in ten families refusing any organ donation for transplantation [29].Our findings emphasize the importance of engaging family members in decision-making about organ donation.

## Study strengths and limitations

To the best of our knowledge, this is the first study in Uganda to investigate the willingness of caregivers for patients with ESRD to donate a kidney. We studied the reasons for and against willingness to donate a kidney among caregivers. Our study sets the ground for further research on willingness for kidney donation in Uganda and elsewhere in the sub-Saharan African region. Our study has some weaknesses. We measured willingness to donate a kidney and this represents an intention and not the practice which involves the actual donation. The study population was from an urban setting and therefore may not represent perspectives from a rural setting.

## Conclusions

Our study shows that more than half of the caregivers for patients with ESRD are willing to donate a kidney for transplantation. Willingness is more likely among older caregivers compared to the younger caregivers, among females compared to males, when permissible by the caregiver's religion, and when the caretaker has a positive attitude towards kidney donation. The barriers to willingness to donate a kidney to a patient with ESRD includes health systems concerns like the lack of confidence in healthcare provided to patients and fear that the surgery might be less successful; family concerns like negative influences from members; religious concerns especially the none acceptance of organ donation; and, personal factors such as loss of body image, death, and prolonged hospitalization that potentially leads to loss of employment. To improve the willingness of caregivers for patients with ESRD in donating a kidney, the social, religious, and personal barriers to kidney donation should be addressed perhaps through health education.

## Supporting information

**S1 Data. Dataset.**
(DTA)

## Acknowledgments

We thank all the caregivers for patients with ESRD for providing data and CIU-REC for reviewing the protocol. This scholarly work is from the MPH dissertation for Hope Bunori (HB).

## Author Contributions

**Conceptualization:** Hope Bunori, Jonathan Izudi, John Bosco Alege.

**Data curation:** Hope Bunori, Jonathan Izudi, John Bosco Alege.

**Formal analysis:** Jonathan Izudi, Francis Bajunirwe.

**Investigation:** Hope Bunori, Jonathan Izudi, John Bosco Alege, Francis Bajunirwe.

**Methodology:** Jonathan Izudi, Francis Bajunirwe.

**Project administration:** Hope Bunori.

**Software:** Jonathan Izudi, Francis Bajunirwe.

**Supervision:** Francis Bajunirwe.

**Validation:** Jonathan Izudi, Francis Bajunirwe.

**Visualization:** Jonathan Izudi, Francis Bajunirwe.

**Writing – original draft:** Hope Bunori, Jonathan Izudi, Francis Bajunirwe.

**Writing – review & editing:** Hope Bunori, Jonathan Izudi, Francis Bajunirwe.

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
