## [Decision Letter · Decision Letter 0]

10 Feb 2022

PGPH-D-22-00048

Willingness of caregivers to donate a kidney to a patient with end-stage renal disease: findings from four dialysis providing health facilities in Uganda

Dear Dr. Jonathan Izudi,

Thank you for submitting your manuscript to PLOS Global Public Health. After careful consideration, we feel that it has merit but does not fully meet PLOS Global Public Health’s publication criteria as it currently stands. Therefore, we invite you to submit a revised version of the manuscript that addresses the points raised during the review process.

We look forward to receiving your revised manuscript.

Kind regards,

Collins Otieno Asweto, PhD

Academic Editor

Journal Requirements:

1. Your manuscript is missing the following sections: Introduction. Please ensure these are present, and in the correct order, and that any references to subheadings in your main text are correct. An outline of the required sections can be consulted in our submission guidelines here: https://journals.plos.org/globalpublichealth/s/submission-guidelines#loc-parts-of-a-submission

Reviewers' comments:

Reviewer's Responses to Questions

**Comments to the Author**

1. Does this manuscript meet PLOS Global Public Health’s publication criteria? Is the manuscript technically sound, and do the data support the conclusions? The manuscript must describe methodologically and ethically rigorous research with conclusions that are appropriately drawn based on the data presented.

Reviewer #1: Yes

Reviewer #2: Yes

2. Has the statistical analysis been performed appropriately and rigorously?

Reviewer #1: Yes

Reviewer #2: Yes

3. Have the authors made all data underlying the findings in their manuscript fully available (please refer to the Data Availability Statement at the start of the manuscript PDF file)?

Reviewer #1: Yes

Reviewer #2: Yes

4. Is the manuscript presented in an intelligible fashion and written in standard English?

Reviewer #1: Yes

Reviewer #2: Yes

5. Review Comments to the Author

Reviewer #1: The title is good and it reflects on the context of the paper. The background information provided is adequate. The methodology is good as well but however, no information was provided on how the participants were selected across the four health facilities used in the study. Sample size was given but no adequate information on sample size computation, such as the formula used to calculate the sample size was not included. The discussion and conclusion are good but more details on the barriers to kidney donation by health care giver can be added.

Reviewer #2: The chosen theme stands out for addressing a pathology of high worldwide prevalence, end-stage renal failure, in which the therapeutic possibilities are complex and the available treatment in which there is a gain in quality of life is kidney transplantation. The need for kidney donors is growing and still faces numerous barriers and the manuscript is right in bringing the facilitators and difficulties that must be addressed in the community to increase adherence to donations. The research is unprecedented in Uganda according to its authors and brings data from several countries on the subject, enriching the discussion, bringing elements of a broader personal, family and social nature, including legal issues.

The study adhered to the Strengthening the Reporting of Observational Studies in Epidemiology (STROBE) guideline and uses a questionnaire that was previously improved, with the aim of qualifying its internal consistency, both in the language approach and its psychometric property.

An element to be considered is the economy and culture of caregivers in Uganda, if it is done only by family and friends or if there are other population groups that are involved in this type of action, such as health workers, if they are paid or not, allowing better comparison with other countries.

The epidemiological analysis is thorough, but the methods do not clarify whether the sample was selected randomly or intentionally and how the sample was selected.

6. PLOS authors have the option to publish the peer review history of their article (what does this mean?). If published, this will include your full peer review and any attached files.

**Do you want your identity to be public for this peer review?** For information about this choice, including consent withdrawal, please see our Privacy Policy.

Reviewer #1: **Yes: **Dr Adebayo Mustapha

Reviewer #2: **Yes: **BOBEK, PAULO RICARDO

---

## [Editor Report · Decision Letter 1]

2 Mar 2022

Willingness of caregivers to donate a kidney to a patient with end-stage renal disease: findings from four dialysis providing health facilities in Uganda

PGPH-D-22-00048R1

Dear Dr. Jonathan Izudi,

We are pleased to inform you that your manuscript 'Willingness of caregivers to donate a kidney to a patient with end-stage renal disease: findings from four dialysis providing health facilities in Uganda' has been provisionally accepted for publication in PLOS Global Public Health.

Best regards,

Collins Otieno Asweto, PhD

Academic Editor
